# TeleRehabilitation of Social-Pragmatic Skills in Children with Autism Spectrum Disorder: A Principal Component Analysis

**DOI:** 10.3390/ijerph20043486

**Published:** 2023-02-16

**Authors:** Flavia Marino, Chiara Failla, Roberta Bruschetta, Noemi Vetrano, Ileana Scarcella, Germana Doria, Paola Chilà, Roberta Minutoli, David Vagni, Gennaro Tartarisco, Antonio Cerasa, Giovanni Pioggia

**Affiliations:** 1Institute for Biomedical Research and Innovation (IRIB), National Research Council of Italy (CNR), 98164 Messina, Italy; 2Classical Linguistic Studies and Education Department, Kore University of Enna, 94100 Enna, Italy; 3Department of Engineering, Università Campus Bio-Medico di Roma, Via Alvaro del Portillo 21, 00128 Rome, Italy; 4Department of Cognitive, Psychological and Pedagogical Sciences, and Cultural Studies, University of Messina, Via Concezione, 6/8, 98121 Messina, Italy; 5Faculty of Psychology, International Telematic University Uninettuno, Corso Vittorio Emanuele II, 39, 00186 Roma, Italy; 6S’Anna Institute, 88900 Crotone, Italy; 7Pharmacotechnology Documentation and Transfer Unit, Preclinical and Translational Pharmacology, Department of Pharmacy, Health Science and Nutrition, University of Calabria, 87036 Arcavacata, Italy

**Keywords:** autism, telehealth, social-pragmatic skills, principal component analysis

## Abstract

In many therapeutic settings, remote health services are becoming increasingly a viable strategy for behavior management interventions in children with autism spectrum disorder (ASD). However, there is a paucity of tools for recovering social-pragmatic skills. In this study, we sought to demonstrate the effectiveness of a new online behavioral training, comparing the performance of an ASD group carrying out an online treatment (n°8) with respect to a control group of demographically-/clinically matched ASD children (n°8) engaged in a traditional in-presence intervention (face-to-face). After a 4-month behavioral treatment, the pragmatic skills language (APL test) abilities detected in the experimental group were almost similar to the control group. However, principal component analysis (PCA) demonstrated that the overall improvement in socio-pragmatic skills was higher for ASD children who underwent in-presence training. In fact, dimensions defined by merging APL subscale scores are clearly separated in ASD children who underwent in-presence training with respect to those performing the online approach. Our findings support the effectiveness of remote healthcare systems in managing the social skills of children with ASD, but more approaches and resources are required to enhance remote services.

## 1. Introduction

After the new era of pandemic emergencies, telehealth systems have become a fundamental service for pursuing and maintaining healthcare protocols. In neurological diseases, such as Parkinson’s [1,2]; Alzheimer’s [3,4] and Multiple Sclerosis [5,6], a large number of tools and services have been proposed.

A clinical domain where the definition of new remote health assistance is mandatory refers to children with autism spectrum disorders (ASD). Indeed, in ASD the need for developing this kind of assistance emerged only after the first COVID-19 pandemic waves. In the last few years, ASD therapy delivery methods using telehealth have been researched, firstly looking at how these technologies might enhance diagnostic testing by providing strategies to assist families in gathering clinically pertinent movies at home and distributing them to diagnostic specialists using telehealth tools [7,8]. Moreover, several authors have demonstrated the effectiveness of telehealth approaches for promoting parent training programs [9,10,11], reducing behavior problems [12] and teaching communication skills [13], and reporting better cognitive and behavioral performance after telehealth protocols, similar to that reached during in-person treatments. Moreover, parents of children with ASD report positive feelings of empowerment [14] and high satisfaction with the use of telemedicine [15,16].

The vast majority of protocols translated on telematic systems with subjects with ASD concern the enhancement of specific domains: social skills [16,17,18,19], parent training [9,19,20], verbal communication [21], and behavioral problems [12,22]. On the other hand, the rehabilitation of social-pragmatic skills has been sparsely translated into telehealth protocols. Pragmatics is widely considered the domain that is distinctively and uniformly lacking in ASD. Problems with pragmatic language (i.e., the appropriate social use of language) continue in ASD even when structural language appears to be intact. A pragmatic language impairment is characterized by a mismatch between the language used and the context in which it is used, such that it is somehow unsuitable for the needs of the circumstance [23]. Some promising approaches have been proposed to treat these deficits, demonstrating that the active involvement of the child and parent in the intervention was found to be a key mediator of the intervention effect [24]. The vast majority of previous protocols have been validated for in-presence treatments. For this reason, a remote assistance protocol focused on this specific deficit is mandatory.

For this reason, this study is aimed at validating a new telehealth program for the recovery of social-pragmatic skills in ASD using an RCT approach, comparing the performance of an ASD group carrying out an online treatment with respect to a control group of ASD children engaged in a traditional in-presence intervention (face-to-face). The current study adds to this literature with the employment of a new remote online tool for improving social-pragmatic abilities in ASD children and with a comparison of behavioral performance analyzed by means of Principal Component Analysis (PCA) for increasing the interpretability and enabling the visualization of data in a new multidimensional way.

## 2. Materials and Methods

### 2.1. Enrollment

Children were recruited and tested at the clinical facilities of the Institute for Biomedical Research and Innovation of the National Research Council of Italy (IRIB-CNR) in Messina. The inclusion criteria were as follows: (1) between 8 and 13 years of age; (2) clinical diagnosis of ASD based on the DSM-5 criteria from a licensed clinical child neuropsychiatrist; (3) a verbal and performance Intelligence Quotient above 75 [25], as assessed by the Wechsler Intelligence Scale for Children-4th edition (WISC-IV); (4) no current problems with aggressive behavior or severe oppositional tendency; (5) no hearing, visual, or physical disabilities that would prevent participation in the intervention; (6) not being on psychiatric medication. All participants have a previous diagnosis that was further confirmed through the assessment and the consensus of experienced professionals on the research team (i.e., a child neuropsychiatrist and a clinical psychologist).

From an initial sample of 42 ASD children, *n* = 26 were excluded because they did not meet the study inclusion criteria. Sixteen children fully met the admission criteria and were enrolled in the present study. Experimental and control groups were pair matched according to demographic variables. All children completed all phases of the rehabilitation protocols and were included in the statistical analysis (Figure 1).

### 2.2. Study Design

The objective of the study was to enhance and develop pragmatic skills in children, diagnosed with ASD, aged 8 to 13 years. A single-blind, randomized controlled study was conducted as part of an ongoing research program and tested at our clinical facilities within the Project INTER PARES “Inclusione, Tecnologie e Rete: un Progetto per l’Autismo fra Ricerca, E-health e Sociale”—POC Metro 2014–2020, Municipality of Messina, ME 1.3.1.b, CUP F49J18000370006, CIG 7828294093. The study’s recruiting of ASD children was the focus of the first phase. Then, at baseline (T0), eligible people received a clinical evaluation. Using a computer-generated randomization code, participants were split into two groups at random in the third stage. The physicians (who conducted the clinical baseline assessment (T0) and post-treatment examination (T1)), were blinded to the parents’ group affiliation. Participants in the experimental group completed a Web intervention of social pragmatic skills, whereas children assigned to the control group underwent the same protocol in a face-to-face approach. At the end of treatment, participants from both groups were given a final evaluation (T1), using the same protocol as the baseline. Four therapists were involved in the different phases of treatment with distinct roles. One therapist (C.F.) was responsible for implementing the protocol in the online setting, another for in-person therapy (P.C.), and two others (R.M., N.V.) for conducting blind evaluations before and after treatment.

### 2.3. Treatment

Following an Italian well-validated protocol for ASD children [26], the treatment was thought to increase the ability to understand the speaker’s intentions and the use of non-literal linguistic expressions. ASD children were trained on 13 specific categories (Table 1) during two sessions characterized by two video training. Each video is made up of drawings of animated animals as if they were cartoons. Two questions are asked at the end of each video. The first is used to understand whether the child has correctly interpreted the general meaning of the story; the second is used to determine the degree to which the child is aware that the expression must be intense in a nonliteral sense. 

The training sessions were carried out following a rigid order (Table 2). In summary, the first step is devoted to improving group social skills. The beginning of the session was devoted to greetings among participants and social interest questions. During the second step, videos were administered and participants answered the target questions, in order to understand if the child has correctly interpreted the general meaning of the story, and to determine the degree to which the child is aware that the expression must be understood in a non-literal sense. In the third step, a role-play was performed among the participants by re-enacting the situations seen in the videos by having all group members take turns participating. During the last step, participants were prompted to share experiences from their lives in which they used or could use the pragmatic skill just analyzed. For the experimental group involved in the telehealth protocol, the training sessions were similar. The online protocol was carried out via a web platform [G-Suite; Google LLC; Mountain View, CA, USA] that provided access to video conferencing tools. The protocol sessions were carried out by children and therapists via videoconference twice a week. Two face-to-face meetings were held during the pre and post-intervention phases to assess socio-pragmatic skills.

### 2.4. Protocol Phases 

The experimental protocol consisted of a total of fifteen phases (see Table 3) divided into 31 sessions/meetings. The training phases lasted 45 min each, twice a week. 

During phase 0, a meeting was organized with the families, where the therapists explained the research objectives and collected consent to participate in the study. At this stage, parents were informed that if their children met the inclusion criteria for the study, two ways of working would be formed: face-to-face treatment and online group treatment.

During phase 1, two meetings were held to assess the subjects with ASD by administering an intelligence test (WISC) to obtain information on the overall level of functioning and one specific test to evaluate the specific difficulty in the sociopragmatic language (APL-MEDEA). At the end of the assessment phase, the subjects were divided into two groups, face-to-face treatment and online treatment. In particular, two groups of four subjects carried out the classic face-to-face training, and two groups of four subjects carried out the treatment online together with a therapist.

From phase 2 to phase 14 ASD children underwent the 13 training phases described above. 

During the last step, phase 15, two blind therapists performed a new evaluation of cognitive performance using the same battery employed at baseline. 

### 2.5. Training Experience and Setting

The therapists who delivered the interventions were all chartered psychologists, or psychotherapists, with behavioral analyst training and at least 5 years of experience in working with ASD children.

The setting for in-person therapy sessions was characterized by a therapy room equipped with a television that is used to broadcast videos related to each phase of the therapy protocol. This provides a dedicated space for participants to focus on their therapy and receive the necessary support and guidance. For online therapy, parents of participants are encouraged to place their children in a room without any distractions. This means creating an environment that is free from noise, other people, and televisions. This quiet setting helps the child to concentrate on their therapy, ensuring that they receive the full benefits of the therapy sessions. Prior to the beginning of the protocol, the therapist and parent established home settings (such as the family room, kitchen, and bedroom) where the laptop may be put for the best viewing. The therapist could then watch a variety of parent-child behaviors and interactions without having to stop what they were doing to ask the parent to adjust the monitor.

### 2.6. Outcome Measurements

The only outcome measure was collected through the administration of a standardized language pragmatics test (APL-MEDEA). Pragmatic Skills Language (APL Medea) [27]) is a pragmatic language skills assessment battery measuring the ability to communicate effectively, taking into account the context, the communicative situation, and the interlocutor’s knowledge. The battery was designed to investigate how pragmatic language skills develop and progress between the ages of 5 and 14.

The battery consists of five sub-tests:Metaphors (M), divided into Verbal Metaphors (MV) and Figured Metaphors (MF): investigate the ability to understand metaphorical language.Understanding of implied meaning (CSI): evaluates the ability to draw inferences on non-explicit content.Comics (C): evaluates the ability to understand and respect the dialogic structure in communication.Situations (S): evaluates the ability to understand and embrace the meaning assumed by particular expressions in social interaction.The color game (CG): evaluates the ability to use language in a referential way and to use skills related to the “Theory of Mind”.

This battery is useful for providing a quantitative assessment of pragmatic skills in understanding and using verbal language.

### 2.7. Statistical Analysis

Univariate analyses were performed using R Statistical Software (Version 4.1.2—R Core Team; https://www.R-project.org/; access date: 1 October 2022). Firstly, the pragmatic language abilities of the two groups were compared at time T0 in order to exclude significant differences before the treatment. 

A statistical power analysis was performed for the estimation of the sample size (G*Power, 3.1; https://www.psychologie.hhu.de (accessed on 1 January 2022). The analysis was based on our clinical experience together with the lack of similar studies in the literature. To estimate the variables for computing power, we considered (1) the average score for typical children from the APL-MEDEA standardization; (2) the average score for children with ASD and normal intelligence from our database; (3) the average age of the participants and; (4) the average differences between ASD and typical participants on the five subscales; (5) we calculated a ‘best case’ scenario with post-training scores in the normal range and a ‘fair case’ scenario with post-training scores at the midpoint between typical and pre-training ASD scores. The average difference between scales is expected to be 4.4 points. We should expect a best-case gain of 4.4 points and a minimum gain of 2.2 points to consider the training worthwhile. Due to the standardization of the measures and the homogeneity of the sample, we expected an average standard deviation of the subscores of 3.1. We should also adjust the alpha value using the Bonferroni correction; therefore, with alpha’ = 0.001 and power = 0.8, the estimated sample size needed for this effect size is approximately *n* = 18 for each group in the best case (expected effect size d = 1.4) and *n* = 68 for each group in the fair scenario (expected effect size d = 0.7). Thus, our initial sample of *n* = 21 for each group would only have been sufficient in the best-case scenario, yet the sample was underpowered after applying the selection criteria.

For this reason, the non-parametric Mann–Whitney U test was employed. Then, the non-parametric Wilcoxon signed-rank test was used to evaluate differences between time T0 and time T1 separately for each of the two groups in order to assess the efficacy of the treatment. All statistical analyses were two-tailed; the statistical threshold was corrected according to Bonferroni: *p* < 0.05/5 = 0.01, considering the five comparisons (APL clinical sub-tests).

Finally, Principal Component Analysis (PCA) was used to reduce the multiple dimensionalities of all the variables related to the assessment of socio-pragmatic skills. The first and second principal components were used for performing scatter plots and observing the separation of treatment in the two groups (Experimental vs. Control) before and after training.

## 3. Results

No significant differences in demographic, IQ values and pragmatic skills at the time of enrollment were detected between the experimental and control groups (Table 4).

Behavioral treatments for socio-pragmatic abilities carried out better performance in both groups (Table 5). In particular, we detected significant increments for ASD children in-presence (control group) in all the parameters except for APL-C and APL-S, while for subjects undergoing online training (experimental group), significant improvements were found only for APL-M (Table 6).

Direct comparisons between the experimental and control groups at follow-up indicated a similar improvement in pragmatic ability, without any predominant main effect of the group (Table 7). Despite no significant differences between groups being detected, it is possible to highlight a general better trend emerging in ASD children who underwent the in-presence treatment (Figure 2).

To better understand the nature of behavioral improvement stimulated by training of pragmatic abilities and how the different modalities (online vs. in-presence approach) impact ASD children, a Principal Component Analysis was performed on the data recorded at baseline versus follow-up. Results of the principal component analysis (Figure 3) demonstrated that the overall improvement in socio-pragmatic skills is higher for ASD children who underwent in-presence training. In fact, the two clusters face-to-face pre-training and face-to-face post-training are separated, while the two clusters referring to tele pre-training and tele post-training are partially overlapped. 

## 4. Discussion

In this study, we demonstrated that our telehealth service is as efficacious as in-person usual care to improve social-pragmatic skills in children with ASD. After a 4-month behavioral treatment, the cognitive abilities detected in the experimental group were almost similar to a group of demographically and clinically matched ASD children followed in-presence. No significant differences were detected in the ability to communicate effectively, taking into account the metaphors (APL-M), the inferences on non-explicit content (CSI), the dialogic structure in a communication (C), the meaning assumed by particular expressions in social interaction (Situations) and the ability to use skills related to the “Theory of Mind” (CG).

Despite early results from univariate investigations, the main conclusion of this study is based on information from PCA. This statistical analysis shows that the area of variability linked with APL performance after treatment was farther out in the in-person group compared to the group that received remote assistance, despite a similar trend being found at follow-up between groups. In fact, the PCI study found overlap in the enhanced performance in the latter group. This result implies that face-to-face rehabilitation of pragmatic abilities is possible to have a bigger impact on the end performance relative to baseline, despite an apparent similar improvement prompted by the two alternative techniques. Instead, developing pragmatic skills through telehealth technology led to a noticeable improvement in behavior; however, the final variability found across all pragmatic sub-scales may have overlapped baseline performance.

Several studies in the literature demonstrated the efficacy and validity of online treatments for recovering behavioral skills in ASD. The majority of studies in the field of study used telemedicine to provide Parent-Training interventions [9,19,20,28], functional life skills training [29], and social skills training for parents or teachers of ASD children [16,17,18,19,30,31]. Lindgren et al. [12] conducted an RCT study to demonstrate the effectiveness of an online treatment delivered to parents of children with ASD by comparing the results to a traditional intervention (face-to-face). These authors found that there were no statistically significant differences in behavioral outcomes between the different approaches. Vismara and colleagues performed a series of telehealth training programs for parents of children with ASD. In the first preliminary study, they used live video conferencing and a self-guided website for teaching parents to implement autism-specific interventions for verbal language and joint attention initiations [21]. Significant improvements were detected in verbal skills but not in attention. In another larger RCT study, these authors found high fidelity in parents when examined 3 months later at follow-up, although children’s social communication did not improve as a result of treatment. Up to now, telehealth is currently regarded as a practical, popular, affordable, and available option for bringing together specialists working with people with ASD [29], although the magnitude of behavioral improvements resulting from remote intervention remains to be defined yet. The employment of different statistical approaches, such as PCA, could aid in increasing the interpretability and enabling the visualization of behavioral data in a new multidimensional way. 

This study has two specific innovations. First, to the best of our knowledge, this is the first RCT study aimed at validating the application of a remote social-pragmatic skills training protocol applied directly to ASD children. Indeed, previous studies [29,30,31] demonstrated the effectiveness of telehealth intervention on social skills but mediated by parents, who were taught the principles of behavior analysis and the implementation of interventions that target functional living skills [29]. Similar to previous evidence, also in this study we confirm that the telehealth approach is effective, as an in-presence modality, to induce evident improvements in social-pragmatic skills. However, the two strategies—tele and in-person—have differing effects on the performance as a whole. Our study is, in fact, the first to use PCA analysis to more thoroughly assess behavioral performance brought on by treatment. We discovered that the in-person rehabilitation strategy is still the most effective way to cause a distinct difference between baseline and follow-up performance reports.

### Limitations

The first limitation of this study was the limited number of participants. A larger sample could provide a more accurate analysis of the effects of treatment. However, the employment of demographically and clinically matched groups, as well as the employment of an RCT approach highlights the importance of our findings. Given the dearth of online behavioral treatments for recovering social skills in ASD children, this constraint is still also a strength. In fact, we feel that we have demonstrated the viability of this kind of training, and we anticipate that this will encourage a wider range of subsequent research that does not solely concentrate on one technique. Other possible limitations refer to the lack of a satisfaction questionnaire in relation to the online service, and a cost analysis to compare the cost of providing traditional face-to-face treatment in situ versus via telehealth.

## 5. Conclusions

In the post-pandemic era, telehealth networks have emerged as a crucial service for pursuing and upholding healthcare regulations. New remote assistance services for children with ASD are defined as an emerging topic of study that is critically needed to meet therapeutic demands. Remote health services can offer a long-term framework for carrying out assessments and educating healthcare workers and personnel on how to apply different behavioral methods. Based on the results of this study, we contend that training sessions for kids with ASD can be carried out using a straightforward remote service while retaining the effectiveness of conventional therapy. To achieve the benefit offered by a face-to-face approach, however, data from PCA analysis reveals that other strategies and tools for improving remote services need to be put into place.

## Figures and Tables

**Figure 1 ijerph-20-03486-f001:**
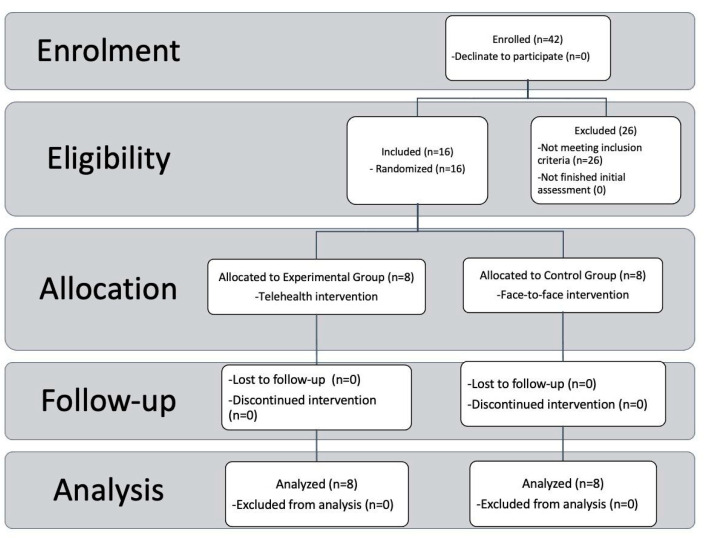
Subject recruitment assignment, and assignment procedures. CONSORT Flow diagram showing the phases of a parallel randomized trial of two groups of ASD individuals underwent telehealth (experimental group) or traditional (control) interventions.

**Figure 2 ijerph-20-03486-f002:**
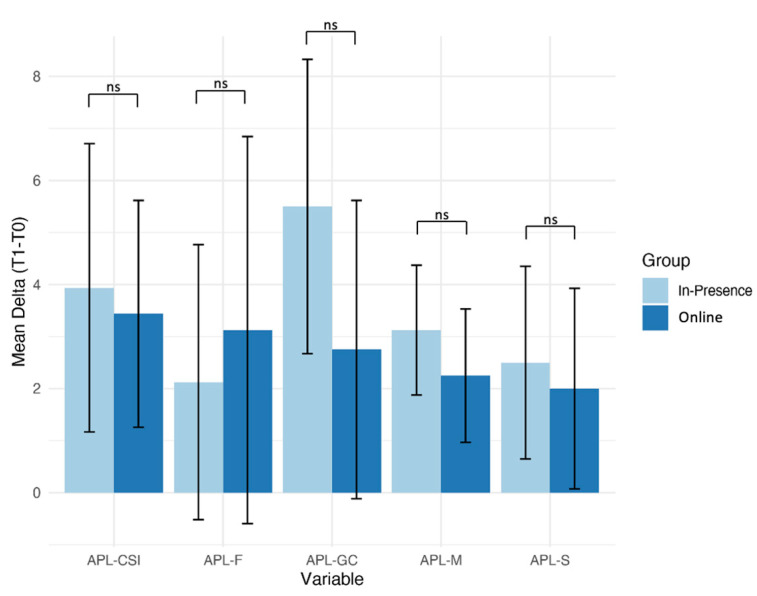
Mean Delta values (T1−T0) for each APL subtests in the two groups. Behavioral changes before and after treatment were calculated as the delta-values. Online training: Experimental group; In-presence: Control group. Error bars represent the SD. ns: no significant.

**Figure 3 ijerph-20-03486-f003:**
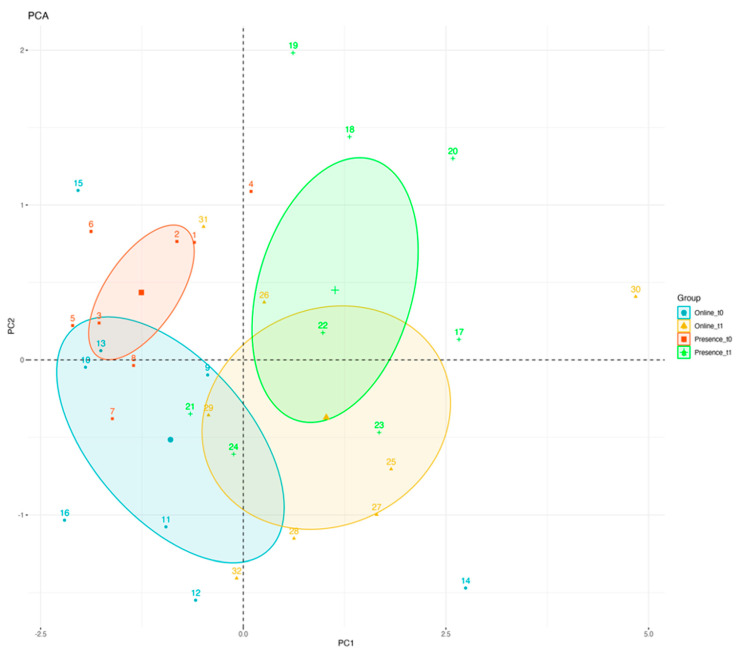
Principal Component Analysis Results. Principal component analysis of behavioral improvements in socio-pragmatic skills as detected in the Experimental group (Online training) and Control group (in-presence training).

**Table 1 ijerph-20-03486-t001:** Video Training.

Pragmatic Categories	Video 1	Video 2
1. Pretend	Children see a rabbit playing with a teddy bear, pretending that a wooden cube is a car.	The child sees a monkey pretending to iron using a pack of rice as an iron
2. Lie	Children see a bunny in the classroom with his classmates while he tells a lie about having a puppy at home because the rest of his class tells of having a pet.	The child sees a skit of a monkey who blames the fox for spilling a cup of chocolate instead of him to defend himself from his mother’s reproach.
3. Appearance/Reality	Children see a little monkey pointing to his friend the fox a cloud and say that it is a sheep because the shape of the cloud is the same as a sheep.	Children see a teddy bear who is afraid to go to bed and tells his mom that there is a witch outside her door because the shadow of the plant that is near her room looks like a witch.
4. Courtesy lie	Children see a donkey giving an ugly sweater to his piggy friend. In order not to upset the donkey, the piggy says that the sweater is very beautiful.	Children see grandmother bear giving her granddaughter a pink sweater, but the granddaughter dislikes pink. The granddaughter tells her grandmother that she really likes her gift.
5. Forgetfulness/Distraction	Children see a bunny rushing out of the house in the morning and accidentally taking her husband’s car keys. She says no when her husband asks if she has seen his car keys.	Children see a piggy preparing the bag to go to sea. The piggy forgets the swimsuit on the bed and leaves his room without seeing it. When his mother asks him if he has taken everything he needs to go to sea, the piggy answers yes.
6. Contrary emotion	Children see a wedding scene between a piggy and a little monkey. Among the wedding guests, there is a rabbit who is in love with the monkey. The rabbit, to hide his emotion, tells everyone that he is very happy for the little monkey who is marrying the piggy.	Children see a little piggy teasing a bear because the bear has a very large nose. Piggy’s mother scolds her son even if she thinks that big nose is really a lot of fun.
7. Joke	Children see two bunnies in the classroom near their teacher. The teacher tells the bunny that she is not just a rabbit but a princess because she is very pretty.	Children see two bears enter the monkey house. The house is very messy because the monkeys’ son has just finished playing. When the bears enter the living room, they inquire whether there has recently been an earthquake.
8. Misunderstanding/Double Meaning	Children see a teddy bear and a piggy studying together. The two friends have to solve a difficult math problem, and they do not know how to do it. At one point, the bear says a light bulb went on, and his friend the piggy asks him where this light bulb is.	Children see a donkey and a mouse sitting in the living room. While they are talking about their neighbor monkey, the little donkey says he cannot see him at all, so the mouse suggests the little donkey put on his glasses.
9. Irony/Sarcasm	Children see a donkey with a dirty car. The bear friend when he sees him exclaims: “How clean is your car!”	Children see a little monkey all jeweled up who meets two friends and asks them for their opinion on her appearance. The friends tell her that she is as beautiful as a Christmas tree.
10. Figurative language	Children see a little monkey who longs to have a motorcycle. The little monkey is still too young to ride however he goes to his daddy and asks him to buy a real motorcycle. Daddy tells him “Who put these shackles on your head?”	Children see a very hungry piggy. The piggy while talking to his fox friend says he would like to eat a mountain of chips. The fox thinks of a mountain made of chips and does not understand what the piggy means.
11. Gaffe	Children see a bear meeting a couple of rabbits. The couple of rabbits tell the bear that every Thursday night they go dancing. The bunny, however, says “my husband is a disaster when he dances, he is such a bear.”	Children see a little monkey meeting a rabbit. The monkey tells the rabbit that he and his friend the bear went on a roller coaster. The little monkey says that the little bear was so afraid and for that reason he was really a rabbit.
12. Persuasion	Children see a little monkey giving a banana to a bunny friend. The bunny says she does not like bananas. Mother rabbit tells her daughter that if she does not accept the banana, the monkey will not want to be her friend anymore.	Children see a little piggy sitting at the table who does not want to eat the spinach that their mother has cooked for him. Mother tells him that he will never be strong if he does not eat the spinach.
13. Complex stories	Children see a little donkey and a little bear watching a soccer game on TV. The little donkey says the national team player is a real monster. The little bear replies that she thinks the player is not so ugly.	Children see a little monkey making a smoothie with licorice and vegetables for his little donkey friend. The little monkey asks the donkey if the smoothie was good, and the donkey replies, “Sure it was, it was the end of the world!”.

**Table 2 ijerph-20-03486-t002:** Training sessions.

Step	Activity	Goals
1° Step	Welcome phase and greetings between group members	Increase the social area
2° Step	Discussion of previous phases and administration of the history provided for within the phase	Specific skill training
3° Step	Role play among the participants of the stories viewed	Skill consolidation
4° Step	Identification and sharing among group members of experiences similar to those faced during the administered phase	Generalization in life contexts

**Table 3 ijerph-20-03486-t003:** Protocol structure.

Phases	Intervention	Meeting
Phase 0	Protocol Explanation	1
Phase 1	Pre-Assessment	2/3
Phase 2	Pretend	4/5
Phase 3	Lie	6/7
Phase 4	Appearance/Reality	8/9
Phase 5	Courtesy Lie	10/11
Phase 6	Forgetfulness/Distraction	12/13
Phase 7	Contrary emotion	14/15
Phase 8	Strike	16/17
Phase 9	Misunderstanding/Double meaning	18/19
Phase 10	Irony/Sarcasm	20/21
Phase 11	Figurative Language	22/23
Phase 12	Gaffe	24/25
Phase 13	Persuasion	26/27
Phase 14	Complex stories	28/29
Phase 15	Post-Assessment	30/31

**Table 4 ijerph-20-03486-t004:** Demographic characteristics of children with ASD.

Measure	Control Group (n°8)	Experimental Group (n°8)	*p*-Level (*Chi^2^*/*Mann-Whitney*)
Gender (M/F)	7/1	8/0	0.302
Age (Years)	9.62 ± 1.92 9 (8–13)	9.12 ± 1.46 *8.5* (*8–12*)	0.659
Total IQ (WISC-IV)	94 ± 9.15 92.5 (82–110)	97 ± 11.24 *102.5 (78–110*)	0.493
APL-M	3.75 ± 2.25 3 (2–8)	6.63 ± 4.24 7 (2–14)	0.159
APL-CSI	4.31 ± 1.96 5.25 (2–6.5)	4.88 ± 2.7 5.25 (1–10)	0.957
APL-C	4.75 ± 3.41 3.5 (1–10)	2.5 ± 2.62 2 (0–7)	0.109
APL-S	1 ± 1.19 0.5 (0–3)	1.63 ± 2.07 1 (0–6)	0.659
APL-GC	1.5 ± 2.82 0 (0–8)	1.5 ± 3.85 0 (0–11)	0.701

Data are given as mean values (SD), and median (range). IQ: Intelligent Quotient. Data are expressed as mean ± SD or median (range) values if assumptions of normality are proved or otherwise. M:Metaphor; CSI: Understanding of implied meaning; F: Comics; S: Situations; GC: Color Game.

**Table 5 ijerph-20-03486-t005:** Delta values (T1–T0) of behavioral improvements in the Experimental (Online) and Control (In-presence) groups after treatments.

	Group	N	Mean	Median	SD	Variance	Range	Min	Max
APL-M	Online	8	2.25	2.00	1.28	1.64	4	0	4
	In-Presence	8	3.13	4.00	1.25	1.55	3	1	4
APL-CSI	Online	8	3.44	3.50	2.18	4.75	7.50	0.00	7.50
	In-Presence	8	3.94	3.75	2.77	7.67	7.00	1.00	8.00
APL-F	Online	8	3.13	3.00	3.72	13.84	12	−2	10
	In-Presence	8	2.13	2.00	2.64	6.98	9	−2	7
APL-S	Online	8	2.00	2.00	1.93	3.71	5	0	5
	In-Presence	8	2.50	2.50	1.85	3.43	5	0	5
APL-GC	Online	8	2.75	2.50	2.87	8.21	8	0	8
	In-Presence	8	5.50	5.00	2.83	8.00	7	2	9

**Table 6 ijerph-20-03486-t006:** Paired *t*-test analysis between T0 and T1 values in the two groups.

	* Control *		* Experimental *	
	Statistic	* p *	Statistic	* p *
APL-M	** 0.00 **	** 0.01 **	0.00	** 0.01 **
APL-CSI	** 0.00 **	** 0.01 **	0.00	0.02
APL-F	3.00	0.06	2.50	0.06
APL-S	0.00	0.03	0.00	0.06
APL-GC	** 0.00 **	** 0.01 **	0.00	0.06

Wilcoxon W test (W-values/*p*-level). Bold values represent significant statistical differences.

**Table 7 ijerph-20-03486-t007:** *t*-Test Analysis between Delta values (T1−T0) between groups.

Scores	Statistic	* p *	Effect Size
APL-M	21.00	0.228	0.345
APL-CSI	29.50	0.833	0.078
APL-F	27.50	0.669	0.140
APL-S	27.00	0.627	0.156
APL-GC	15.00	0.080	0.531

Mann-Whitney *t*-test (U-values/*p*-level).

## Data Availability

The datasets generated during the current study are available from the corresponding author on reasonable request.

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
