# Peer review of "TeleRehabilitation of Social-Pragmatic Skills in Children with Autism Spectrum Disorder: A Principal Component Analysis"

_ijerph, 2023, doi:10.3390/ijerph20043486_

Round 1

Reviewer 1 Report

Summary

In this paper Marino et al. evaluate the effectiveness of a treatment for improving socio-pragmatic skills in telerehabilitation modality on a small group of subjects with autism spectrum disorder . In order to assess this they compared the socio-pragmatic skills of a sample of subjects in presence modality with those of a sample treated in telerehabilitation and found comparable improvements. Additional statistical analyses, however, confirm that the improvement is greater for subjects who performed the treatment in presence modality.

Evaluation

The study appears sufficiently well executed and the writing is sometimes and communicates the main points.  Moreover, the topic of the paper could interest a large audience, given the growing interest in telerehabilitation in recent years. I think that some points should be addressed or clarified before the study is suitable for publication. My major concerns are mainly related to the use of PCA with such a small sample of subjects and above all to the absence of details concerning statistical analysis.

Introduction

Here, from my point of view, the authors should explain what socio-pragmatic abilities are and why it is important to treat them in autism spectrum disorders. Citing also existing rehabilitation studies about this topic.

Methods

1.     It would be helpful to know whether the treatment protocol is based on any theoretical models or previously published efficacy studies.

Results

2.     Table 2 or in the text the authors should report, following APA style, full statistics details ( such as degree of freedom, effect size), in particular giving the small sample size in this case effect size is really important to support the effectiveness of treatment.  In particular for the result regarding table 2, the appropriate statistical test should be a two-way repeated measures Anova, with planned t-test to compare subtests of APL-Medea.

3.     In figure 1 one the authors should explain in the caption if the error bars are SEM or SD.

were there any subjects who did not respond, or respond better,  to treatment?

4.     For the PCA I am really think that minimum sample size (5 or 10 cases for each variables ) is not reached, moreover the authors should an extended report of the analysis that support their statements.  In the caption of figure 2 Remote training group is reported as online training and in the legend is reported as tele_t0, please the author to choose a single term to indicate the remote training group, the same for the text. Again for the figure 2 what are the symbols in the graphs, now labeled with numbers?

Are there any clinical variables at t0 that predict response to treatment in the 2 groups at follow-up? 

Author Response

Reviewer n°1

In this paper Marino et al. evaluate the effectiveness of a treatment for improving socio-pragmatic skills in telerehabilitation modality on a small group of subjects with autism spectrum disorder . In order to assess this they compared the socio-pragmatic skills of a sample of subjects in presence modality with those of a sample treated in telerehabilitation and found comparable improvements. Additional statistical analyses, however, confirm that the improvement is greater for subjects who performed the treatment in presence modality.

The study appears sufficiently well executed and the writing is sometimes and communicates the main points.  Moreover, the topic of the paper could interest a large audience, given the growing interest in telerehabilitation in recent years. I think that some points should be addressed or clarified before the study is suitable for publication. My major concerns are mainly related to the use of PCA with such a small sample of subjects and above all to the absence of details concerning statistical analysis.

Introduction

  • Here, from my point of view, the authors should explain what socio-pragmatic abilities are and why it is important to treat them in autism spectrum disorders. Citing also existing rehabilitation studies about this topic.

REPLY: Following reviewer’s suggestion the introduction has been improved.

 Methods

  1. It would be helpful to know whether the treatment protocol is based on any theoretical models or previously published efficacy studies.

REPLY: We followed the well-validated protocol for socio-pragmatic abilities made by “Rosati, S., & Urbinati, N. (2016). Allenare le abilità socio-pragmatiche. Storie illustrate per bambini con disturbi dello spettro autistico e altri deficit di comunicazione. Trento: Ed. Erickson. ISBN: 9788859010616”. This is a well established protocol used in several clinical units for the treatment of deficits in social communication and abilities of children with ASD.

Results

  1. Table 2 or in the text the authors should report, following APA style, full statistics details ( such as degree of freedom, effect size), in particular giving the small sample size in this case effect size is really important to support the effectiveness of treatment.  In particular for the result regarding table 2, the appropriate statistical test should be a two-way repeated measures Anova, with planned t-test to compare subtests of APL-Medea.

REPLY: We thank the reviewer for the useful suggestions. We updated Table 2 with the effect size information and a new table 3 has been included to better show the entire behavioral dataset. For this Analysis we employed the Mann-Whitney test with the aim of determining if there were any significant differences in the pre-post treatment improvement between the experimental group and the control group. We cannot employ parametric methods due to sample size (see now Statistical Section). The improvement was represented by a single variable (delta value) calculated as the difference between the scores at baseline and follow-up.

  1. In figure 1 one the authors should explain in the caption if the error bars are SEM or SD. were there any subjects who did not respond, or respond better,  to treatment?

REPLY: We thank the reviewer for the comment. In figure 1 error bars are SD, we inserted this information in the caption. All individuals enrolled in this study responded to treatment.

  1. For the PCA I am really think that minimum sample size (5 or 10 cases for each variables ) is not reached, moreover the authors should an extended report of the analysis that support their statements.  In the caption of figure 2 Remote training group is reported as online training and in the legend is reported as tele_t0, please the author to choose a single term to indicate the remote training group, the same for the text. Again for the figure 2 what are the symbols in the graphs, now labeled with numbers?

REPLY: We thank the reviewer for the valuable suggestions and questions. Principal Component Analysis was employed only as an instrument of visualization in order to show how data are clustered in the two groups (Control vs Experimental) before and after training, confirming results obtained from previous statistical analysis. Such analysis was performed on original variables and not after dimensionality reduction. However PCA was applied on a dataset with 32 samples (16 subjects per two time points) per variable and both sphericity and sampling adequacy were checked with Barlett and Kaiser-Meyer-Olkin tests respectively. Results of these tests are reported in the following tables:

Bartlett’s test of sphericity

χ²

gdl

p

50.9

10

<.001

p-value is < 0.05 so the assumption is verified.

KMO measure of sampling adequacy

MSA

Global

0.796

APL_M

0.770

APL_CSI

0.762

APL_F

0.831

APL_S

0.808

APL_GC

0.857

Each measure is >0.7 so the assumption is verified.

In Figure 2, each number identifies a different sample in the dataset (32 in total), whereas each symbol represents a different cluster, as reported in the legend on the right (Control Group and Experimental Group before and after training). Moreover we modified figures aligning the label for the experimental group choosing the term “Online”.

4) Are there any clinical variables at t0 that predict response to treatment in the 2 groups at follow-up? 

Reply: the small sample size limited the application of predictive mathematical or statistical models (for instance Machine Learning) which need greater sample size (minimum n*100). For instance see our last paper: Cerasa A, Tartarisco G, Bruschetta R, Ciancarelli I, Morone G, Calabrò RS, Pioggia G, Tonin P, Iosa M. Predicting Outcome in Patients with Brain Injury: Differences between Machine Learning versus Conventional Statistics. Biomedicines. 2022 Sep 13;10(9):2267.

Reviewer 2 Report

#Review

TeleRehabilitation of social-pragmatic skills in Children with autism spectrum disorder: a Principal Component Analysis

Dear Authors,

Thank you very much for the opportunity to revise this fascinating manuscript about the impact of telerehabilitation of social-pragmatic skills in the context of autism spectrum disorder. This work might be considered relevant and innovative, given the importance of telehealth practices nowadays. However, several main concerns should be addressed before the official publication in the International Journal of Environmental Research and Public Health.

My comments are stated below.

Introduction

In general, the introduction is too concise, and it does not provide a clear picture of recent telehealth practices in the context of ASD, which areas they target, and how. Furthermore, children’s age should be mentioned when describing the studies, given its relevance in a developmental perspective.

There are relevant works of Laurie A. Vismara that should be mentioned and described in more detail.

e.g., see 

Vismara, L. A., McCormick, C., Young, G. S., Nadhan, A., & Monlux, K. (2013). Preliminary findings of a telehealth approach to parent training in autism. Journal of autism and developmental disorders, 43(12), 2953-2969.

Vismara, L. A., McCormick, C. E., Wagner, A. L., Monlux, K., Nadhan, A., & Young, G. S. (2018). Telehealth parent training in the Early Start Denver Model: Results from a randomized controlled study. Focus on Autism and Other Developmental Disabilities, 33(2), 67-79.

Vismara, L. A., Young, G. S., & Rogers, S. J. (2012). Telehealth for expanding the reach of early autism training to parents. Autism research and treatment, 2012.

Other relevant works might be:

Smith, C. J., Rozga, A., Matthews, N., Oberleitner, R., Nazneen, N., & Abowd, G. (2017). Investigating the accuracy of a novel telehealth diagnostic approach for autism spectrum disorder. Psychological assessment, 29(3), 245.

Talbott, M. R., Dufek, S., Zwaigenbaum, L., Bryson, S., Brian, J., Smith, I. M., & Rogers, S. J. (2020). Brief report: Preliminary feasibility of the TEDI: A novel parent-administered telehealth assessment for autism spectrum disorder symptoms in the first year of life. Journal of Autism and Developmental Disorders, 50(9), 3432-3439.

and others.

I think the introduction should include more literature and a description of telehealth practices and should be restructured to add more information.

Materials and methods

Enrollment

  • Line72: How was the verbal and performance developmental quotient assessed? Please specify this. 

  • Participants' demographics should be moved here instead of in the results section. The table should follow APA criteria. 

    • a power analysis should be conducted to identify the power of these results, given the paucity of data

Treatment

  • which questions are asked at the end of each video? What are the “social interest questions” asked to participants at the beginning of the session?

  • More information should be added when describing the treatment. How was the laboratory setting during the face-to-face meetings? Are there any devices that should be taken into consideration during the telehealth protocol? (e.g., quiet environment and so on). 

Protocol Phases

  • Phase 1: Please describe the intelligence test (name, reference, description…) and the two specific tests to evaluate the difficulty in sociopragmatic language

  • More information about the therapists should be provided. Did the same therapists apply both face-to-face and telehealth protocols? or did the therapists differ between the two interventions?

Statistical Analysis

  • Specify the choice behind using the non-parametric test. Which tests did the author use to check the variables for normality and linearity?

  • Could the authors justify the choice of applying the Principal Component Analysis with a small sample size?

Discussion

In general the discussion part should be deepened (as the introductory part) with more references on telehealth practice in autism research. Each finding should be appropriately discussed, and the authors should provide justification and support for the findings that used the PCA.

Author Response

REVIEWER n°2

Thank you very much for the opportunity to revise this fascinating manuscript about the impact of telerehabilitation of social-pragmatic skills in the context of autism spectrum disorder. This work might be considered relevant and innovative, given the importance of telehealth practices nowadays. However, several main concerns should be addressed before the official publication in the International Journal of Environmental Research and Public Health.

My comments are stated below.

Introduction

  • In general, the introduction is too concise, and it does not provide a clear picture of recent telehealth practices in the context of ASD, which areas they target, and how. Furthermore, children’s age should be mentioned when describing the studies, given its relevance in a developmental perspective. There are relevant works of Laurie A. Vismara that should be mentioned and described in more detail.
  • Vismara, L. A., McCormick, C., Young, G. S., Nadhan, A., & Monlux, K. (2013). Preliminary findings of a telehealth approach to parent training in autism. Journal of autism and developmental disorders, 43(12), 2953-2969.
  • Vismara, L. A., McCormick, C. E., Wagner, A. L., Monlux, K., Nadhan, A., & Young, G. S. (2018). Telehealth parent training in the Early Start Denver Model: Results from a randomized controlled study. Focus on Autism and Other Developmental Disabilities, 33(2), 67-79.
  • Vismara, L. A., Young, G. S., & Rogers, S. J. (2012). Telehealth for expanding the reach of early autism training to parents. Autism research and treatment, 2012.
  • Smith, C. J., Rozga, A., Matthews, N., Oberleitner, R., Nazneen, N., & Abowd, G. (2017). Investigating the accuracy of a novel telehealth diagnostic approach for autism spectrum disorder. Psychological assessment, 29(3), 245.
  • Talbott, M. R., Dufek, S., Zwaigenbaum, L., Bryson, S., Brian, J., Smith, I. M., & Rogers, S. J. (2020). Brief report: Preliminary feasibility of the TEDI: A novel parent-administered telehealth assessment for autism spectrum disorder symptoms in the first year of life. Journal of Autism and Developmental Disorders, 50(9), 3432-3439.

I think the introduction should include more literature and a description of telehealth practices and should be restructured to add more information.

REPLY: We would like to thank this reviewer for this comment. The introduction has been modified following this suggestion.

Materials and methods

Enrollment

  • Line72: How was the verbal and performance developmental quotient assessed? Please specify this. 

REPLY: The intelligence quotient of children with ASD was assessed using the Wechsler Intelligence Scale for Children–Fourth Edition (Wechsler 2003). This standardized test provides a measure of intellectual ability and cognitive processing. Wechsler, D. (2003). Wechsler intelligence scale for children-fourth edition (WISC-IV). San Antonio, TX: Psychological Corporation. CITAZIONE WISC-IV

  • Participants' demographics should be moved here instead of in the results section. The table should follow APA criteria. 

REPLY: Done

  • a power analysis should be conducted to identify the power of these results, given the paucity of data

REPLY: A power analysis has been included. See the statistical Section

Treatment

  • which questions are asked at the end of each video? What are the “social interest questions” asked to participants at the beginning of the session?

REPLY: At the beginning of each session, during the first step, the participants are left free to interact with each other in order to stimulate and enhance their social initiatives by giving them the opportunity to ask the other group members "What did you do yesterday? ", "Have you finished your homework for today?" etc. This moment of the session aims to increase knowledge among the members of the group. There are two target questions presented at the end of each video and each of them has a specific objective:

  • firstly, to understand if the child has correctly interpreted the general meaning of the story;
  • and to determine the degree to which the child is aware that the expression must be understood in a non-literal sense.

This additional information has been included in the Methods Section.

  • More information should be added when describing the treatment. How was the laboratory setting during the face-to-face meetings? Are there any devices that should be taken into consideration during the telehealth protocol? (e.g., quiet environment and so on). 

REPLY:  Following reviewer’s suggestion we included this additional statement

“The setting for in-person therapy sessions is characterized by a therapy room equipped with a television that is used to broadcast videos related to each phase of the therapy protocol. This provides a dedicated space for participants to focus on their therapy and to receive the necessary support and guidance. For online therapy, parents of participants are encouraged to place their children in a room without any distractions. This means creating an environment that is free from noises, other people, and televisions. This quiet setting helps the child to concentrate on their therapy, ensuring that they receive the full benefits of the therapy sessions. Prior to the beginning of protocol, the therapist and parent established home settings (such as the family room, kitchen, and bedroom) where the laptop may be put for the best viewing. The therapist could then watch a variety of parent-child behaviors and interactions without having to stop what they were doing to ask the parent to adjust the monitor.”

Protocol Phases

  • Phase 1: Please describe the intelligence test (name, reference, description…) and the two specific tests to evaluate the difficulty in sociopragmatic language

REPLY: Following the reviewer’s suggestion we better describe this outcome measurement. During phase 1, two evaluations were held on subjects with ASD by administering an intelligence test and a specific test to assess their language skills. The Wechsler Intelligence Scale for Children–Fourth Edition was used to measure overall intellectual ability and obtain a general IQ score for screening purposes. The Pragmatic Skills Language (APL Medea) test was used to evaluate the subjects' ability to effectively communicate, including their use of non-literal language.

  • More information about the therapists should be provided. Did the same therapists apply both face-to-face and telehealth protocols? or did the therapists differ between the two interventions?

REPLY:  Four therapists were involved in the different phases of treatment with distinct roles. One therapist was responsible for implementing the protocol in the online setting, another for in-person therapy, and two others for conducting blind evaluations before and after treatment. This additional information has been included in the main text

Statistical Analysis

  • Specify the choice behind using the non-parametric test. Which tests did the author use to check the variables for normality and linearity?

REPLY: Assumptions for normality test (Kolmogorov–Smirnov) and Power analysis (see statistical section) confirmed the employment of a non-parametric approach.

9) Could the authors justify the choice of applying the Principal Component Analysis with a small sample size?

            REPLY: We thank the reviewer for the valuable suggestions and questions.

Principal Component Analysis was employed only as an instrument of visualization in order to show how data are clustered in the two groups (Control vs Experimental) before and after training, confirming results obtained from previous statistical analysis. Such analysis was performed on original variables and not after dimensionality reduction. However PCA was applied on a dataset with 32 samples (16 subjects per two time points) per variable and both sphericity and sampling adequacy were checked with Barlett and Kaiser-Meyer-Olkin tests respectively. Results of these tests are reported in the following tables:

Bartlett’s test of sphericity

χ²

gdl

p

50.9

10

<.001

p-value is < 0.05 so the assumption is verified.

KMO measure of sampling adequacy

MSA

Global

0.796

APL_M

0.770

APL_CSI

0.762

APL_F

0.831

APL_S

0.808

APL_GC

0.857

Each measure is >0.7 so the assumption is verified.

In Figure 2, each number identifies a different sample in the dataset (32 in total), whereas each symbol represents a different cluster, as reported in the legend on the right (Control Group and Experimental Group before and after training). Moreover we modified figures aligning the label for the experimental group choosing the term “Online”.

Discussion

10) In general the discussion part should be deepened (as the introductory part) with more references on telehealth practice in autism research. Each finding should be appropriately discussed, and the authors should provide justification and support for the findings that used the PCA.

REPLY: We would like to thank this reviewer for this comment. The Discussion has been modified following this suggestion.

Round 2

Reviewer 1 Report

the authors clarified all my concerns 

Reviewer 2 Report

Dear authors, 

Thank you very much for the opportunity to check this revised version of the manuscript. 

The authors addressed and answered appropriately all my doubts and concerns about the manuscript. Therefore, I would recommend this work for publication.